# Research and Development Efficiency in Public and Private Sectors: An Empirical Analysis of EU Countries by Using DEA Methodology

**Martina Halaskova [1]**, **Beata Gavurova [2,*]** and **Kristina Kocisova [3]**

[1] Faculty of Economics, VŠB-Technical University of Ostrava, Sokolská třída 33, 70200 Ostrava, Czech Republic; martina.halaskova@vsb.cz

[2] Center for Applied Economic Research, Faculty of Management and Economics, Tomas Bata University in Zlin, Mostní 5139, 76001 Zlín, Czech Republic

[3] Faculty of Economics, Technical University of Košice, Němcovej 32, 04200 Košice, Slovak Republic; kristina.kocisova@tuke.sk

\* Correspondence: gavurova@utb.cz; Tel.: +421-944-420-654

**Abstract:** Both the fourth industrial revolution (Industry 4.0) and its embedded technology diffusion exponentially progress and grow in terms of technical change and socioeconomic impact. The aim of this study was the evaluation of research and development efficiency in the public and private sectors in EU countries. The Data Envelopment Analysis (DEA) methodology, within which the slack-based model was applied, was used to achieve this aim. The Malmquist index (MI) was used to calculate changes in research and development efficiency during 2010/2013 and 2014/2017. The results present a decrease in total Research and Development (R&D) productivity in public and private sectors for an average of EU countries (28). However, Spain, Slovenia, and Portugal (in the public sector), and Ireland and Romania (in the private sector) revealed an increase of a total R&D productivity during 2010/2013 and 2014/2017 that was primarily influenced by an increase of technical efficiency (catch-up effect). Similarly, the results confirm the differences in R&D efficiency in private and public sectors in the European countries. The study's results also provide a valuable platform for creators of national strategic and innovative investment and educational plans, and creators of relevant policies and create a platform for national and international benchmarking indicators.

**Keywords:** research; development; efficiency; public sector; private sector; DEA; Malmquist index

## 1. Introduction

Research and development (R&D) and innovation are a central area of individual national and international policies and innovative strategy. Principally, it is related to R&D policies' connection with education, innovation, employment, information, and business policy [1,2]. Research and development play a key role in generating new knowledge, products, and technological processes, which are a necessary condition for stable and sustainable social growth. If Europe wants to become a more competitive knowledge-based economy, not only the production but also the spread and use of knowledge need to improve. It is essential to manage use and effective transfer of knowledge among research organisations, universities and public organisations in particular, and industry small- and medium-scale businesses which transform it into products and services [3–7].

The rapid pace of technological developments played a key role in the previous industrial revolutions. However, the fourth industrial revolution (Industry 4.0) and its embedded technology diffusion progress are expected to grow exponentially in terms of technical change and socioeconomic impact [8–10]. "Industry 4.0" is the common term referring to the fourth industrial revolution. However,

academics still struggle to define its approach appropriately. The key promoters of this idea, the Industry 4.0 Working Group and the Platform Industry 4.0, describe the vision, basic technologies this idea aims at, and selected scenarios [11], but they do not provide a clear definition. Consequently, a generally accepted definition of Industry 4.0 has not been published yet [12]. In 2011, the term 'Industry 4.0' became publicly known. At that time, an initiative called Industry 4.0, i.e., an association of representatives from business, politics, and academia, promoted this idea as an approach to strengthen the competitiveness of manufacturing industry in Germany [13]. The German federal government supported this idea by announcing Industry 4.0 as an integral part of its initiative called 'High-Tech Strategy 2020 for Germany', which aims at technology innovation leadership.

Science, technology, and innovation represent a successively larger category of activities which are highly interdependent but also distinct [14–16]. According to Brooks [17], science contributes to technology in different ways: new knowledge, which serves as a direct source of ideas for new technological possibilities; source of tools and techniques for more efficient engineering design and a knowledge base for evaluation of the feasibility of designs; the practice of research as a source for development and assimilation of new human skills and capabilities eventually useful for technology; creation of a knowledge base that becomes increasingly important in the assessment of technology in terms of its wider social and environmental impacts or knowledge base that enables more efficient strategies of applied research, development, and refinement of new technologies.

Europe 2020 is the European strategy for growth and jobs for the current decade. In research and development (R&D), the goal is for member states to reach 3 percent of the EU's gross domestic product to be invested in R&D. However, according to the latest review of the strategy by Eurostat, "For three consecutive years, R&D expenditure in the EU has stagnated around 2.03 percent of GDP, further decreasing the chances that the EU will reach its 3 percent target" (i.e., in the public sector 1% and private sector 2%) [18].

New findings in science and technology, industry development in research and development, and trends in the knowledge-based economy are related to a realisation of public and private sectors within basic research, applied research, and experimental development [19–22]. According to the OECD [22], public research in EU countries includes activities of the government sector and higher-education sector and is mainly connected to basic research. The government sector is connected to public research institutions carrying out R&D in most cases as their major economic activity. The higher-education sector includes R&D workplaces, mainly faculties and other places of public and state-owned universities, teaching hospitals, private universities, and other research institutions of post-secondary education. Public research is, broadly speaking, performed in either higher education institutions or public research-performing organisations. Both of these sectors contain a very diverse range of institutions of different sizes, budgets, and missions [22]. The business enterprise (private) sector plays an important role due to the globalisation process, which introduces new companies and products to national markets, thus increasing business competition. The private sector especially focuses on applied research (industrial research is its part) and experimental development. The results of these activities are related to innovations and/or patenting activity. The R&D of the business enterprise (private) sector includes all resident corporations, including companies incorporated under the laws and all other types of quasi-corporations that would make a profit or any other profit for their owners [22].

As Skrinjaric [23] states, obtaining information from sector disaggregation is important from a perspective of private research (private business sector R&D). Preservation of the industrial base in Europe and its competitiveness with private R&D, and similarly the excess of private R&Ds to higher education and greater adoption of new technologies, are the main reasons.

Some authors deal with a status and a significance of public and private sectors in research and development, a mutual relationship of both sectors, the role of research institutions, influence of public and private resources that support research, development, and innovation activities, or the linkages between technology and public science [2,14,24,25]. Ravselj and Hodzic [26] showed that not much

is known about the role of public governance in promoting research and development (R&D) in the business sector in the EU. The authors aimed to explain the interaction between the public and business sectors in a cross-national setting by investigating the relationship between different public governance practices and business R&D activity.

Innovation is typically a policy where the European added value is felt, as the scale of the EU allows for bigger projects to be funded, experiments to be run at a higher scale, and standards to be applied over a larger territory [27]. Innovation performance and innovation policy are closely linked to the evaluation and efficiency of R&D [7,28,29]. From the point of development concepts, innovation policy in its simplest form is based on a linear understanding of the process of innovation (science-push), which considers innovations as a logical result of successful R&D. Consequently, innovation policy blends with a science-research policy whose critical task is to support R&D [12,30,31]. Janger et al. [32] evaluated the usefulness of the new indicator against the background of the difficulties in measuring innovation outputs and outcomes and concluded that the new indicator is biased towards a somewhat narrowly defined "high-tech" understanding of innovation outcomes. The authors also showed that the results for the modified indicator differ substantially for a number of countries, with potentially wide-ranging consequences for innovation and industrial policies.

Cai [33] examined efficiency scores of the National Innovation System (NIS) for 22 countries, including the BRICS (the economic bloc of countries consisting of Brazil, Russia, India, and China), G7 (consisting of Canada, France, Germany, Italy, Japan, the United Kingdom and the United States), using Data Envelopment Analysis (DEA). In contrast to the composite indicator approach, the DEA approach focuses exactly on the input-output efficiency of innovation systems. The results of the efficiency showed that the BRICS differ greatly in the efficiency of their NISs, with China, India, and Russia ranking fairly high, and Brazil and South Africa ranking low. To improve the efficiency of innovation systems, efforts should be made to improve the market conditions, governance, and financial structures, and create a sound environment for R&D.

Evaluation of R&D efficiency and innovation efficiency is the main topic of numerous European and global researches (e.g., [25,31]). Efficiency is generally described in terms of the cost per unit of production. In research, it may essentially be used to test research performance by measuring whether the ratio of research output/input may be optimised, by comparing, for example, to other research programmes and/or countries (external benchmarking), or to the previous years' performance (internal benchmarking) [34]. Skrinjaric [23] examined the efficiency of 29 select European countries for the period ranging from 2007 to 2017 in achieving and obtaining R&D goals. The author conducted dynamic analysis and tracked changes of (in)efficiencies over time. The decomposition of the efficiency was performed by separating the main variables of interest into the private, higher education, and government sectors, and the robustness of the results was evaluated. Laliene and Sakalas [35] defined the concepts of R&D productivity and efficiency and provided an analysis of existing R&D assessment structures or models as well as identifying its advantages and disadvantages. Ekinci and Karadayi [36] summarised the studies related to R&D efficiencies of countries and compared the efficiencies of the 27 European Union countries with respect to their R&D activities and to measure the relative efficiency scores by using Data Envelopment Analysis.

Other authors, e.g., [37–40], evaluated the relative performances of public-funded research and development (R&D) organisations functioning across multiple countries, working on similar research streams. The relative efficiencies of the organisations were assessed by output variables (external cash flow, and the numbers of technologies transferred, publications and patents) and input variables (number of grants received from the parent body, and the number of scientific personnel working in these public R&D organisations). Beneito, Rochina-Barrachina, Sanchis [5] investigated the pattern of R&D efficiency in terms of the number of product innovations achieved by firms over time and proposed a model that explicitly acknowledges the twofold composition of firms' R&D expenditures, comprising spending on both physical capital for R&D projects and payments to researchers. The authors regarded this latter component of R&D as a source for dynamic returns to firms' R&D investments. Dobrzanski

and Bobowski [41] examined whether funds spent on research and development are used efficiently in 15 countries (Association of Southeast Asian Nations—ASEAN) in the 2000-2016 period. Measuring the efficiency of research and development spending was performed using the non-parametric Data Envelopment Analysis (DEA) methodology, using the constant return to scale approach and the variable return to scale approach. The research used variables as annual public and private spending on innovation, high-technology exports as a percentage of manufactured exports, patent applications to the World Intellectual Property Organisation by priority year for million inhabitants, trademark applications for million inhabitants, and information and communications technology exports as a percentage of manufactured exports. The study confirmed that increased spending on innovation results in non-proportional effects.

Many authors have examined research and development efficiency by using DEA methodology, and they evaluated changes of efficiency in time by using the Malmquist index, which focuses on two components, catch-up and frontier-shift effect [28,39–43]. Hu, Yang, and Chen [28] applied the distance function approach for stochastic frontier analysis (SFA) to compare research and development (R&D) efficiency across 24 nations during 1998–2005. In this multiple input-output framework, R&D expenditure stock and R&D workforce were used as inputs, while patents, scientific journal articles, and royalties and licensing fees (RLF) were used as outputs. Guan et al. [44] examined the influence of collaboration network structure on national research and development (R&D) efficiency and measured R&D efficiency scores by using the Malmquist productivity index associated with data envelopment analysis. The authors provided country-level evidence that the collaboration network structure influences the R&D result performance measured by output quantity. The results reconfirmed that collaboration network structure influences scientific publications at the country level.

Sharma and Thomas [45] examined the relative efficiency of the R&D process across a group of 22 developed and developing countries using Data Envelopment Analysis (DEA). The R&D technical efficiency was examined using a model with patents granted to residents as the output and gross domestic expenditure on R&D and the number of researchers as inputs. Under CRS (Constant Returns to Scale), Japan, the Republic of Korea, and China were found to be efficient, whereas under the VRS (Variable Returns to Scale) framework, Japan, the Republic of Korea, China, India, Slovenia, and Hungary were found to be efficient. The inefficiency in the R&D resource usage indicates the underlying potential that can be tapped for the development and growth of nations.

Wang [39] constructed a cross-country production model for evaluating the relative efficiency of aggregate R&D activities. Stochastic frontier methods incorporating translog specification were applied to the data of 30 countries in recent years. R&D capital stock and manpower were considered as inputs; patents and academic publications were regarded as outputs. R&D performance indices showed a positive correlation with income level. Policy implications on resources allocation and R&D strategies were discussed. Li and Wang [46] examined R&D input-output performance of the major sectors of industrial enterprises based on the DEA method. The authors analysed the major problems of low efficiency of input-output performance of R&D activities and proposed to solve the problems by combining with the current status of R&D activities of industrial enterprises, with the goal to provide references for the improvement of the efficiency of input-output performance of R&D activities of the major sectors of industrial enterprises in Hebei Province.

The aim of this study is the evaluation of research and development efficiency in public and private sectors in EU countries and evaluation changes of efficiency of R&D by using the Malmquist index during 2010/2013 and 2014/2017. As opposed to numerous comparative analyses and research studies that predominantly evaluate the efficiency of R&D as a whole with all sectors, our study evaluates the efficiency of R&D by the sectors' performance (in government, higher education, and business enterprise sectors) by means of the empirical analysis.

Three research questions are verified to achieve the study's aim: RQ1: Are the European countries efficient in the process of transformation of investment into the research and development into the outputs in the form of scientific and citable documents and patens and high-tech export? RQ2: Was R&D

efficiency in the public sector in the European countries during 2010/2013 and 2014/2017 influenced by technological progress? RQ3: Were the changes in R&D efficiency in private sector in the European countries significantly influenced by a technical efficiency during 2010/2013 and 2014/2017?

## 2. Materials and Methods

According to Hinrichs-Krapels and Grant [34], efficiency in research and development (R&D) can essentially be used to test research performance—measuring whether the ratio of output/input of research can be optimised—by comparing, for example, against other research programmes or countries. Technical efficiency R&D: A measure of how well an input (such as researchers in the public/business sector) is converted into an output (scientific publications, citable documents/patents or innovations). They are measured as the ratio of physical output to physical input. The term productivity can be interpreted in a number of ways. As a general concept, R&D productivity growth is the growth in the volume of output (scientific publications/patents or high-tech exports) relative to growth in the volume of inputs (expenditure, researchers by sectors). However, measures of productivity vary in how inputs and outputs are calculated.

### 2.1. Methods

This paper tried to examine efficiency by looking at the relationship between used inputs and produced outputs. Data Envelopment Analysis is a non-parametric method which may deal with multiple inputs and multiple outputs in efficiency evaluation. The basic DEA model was developed by Charnes et al. [47], where the CCR (Charnes, Cooper, Rhodes) model assumes the existence of a constant return to scale. The basic model was modified by Banker et al. [48], where the BCC (Banker, Charnes, Cooper) model assumes the existence of a variable return to scale. Both of these DEA models were constructed in the form of input- and output-oriented models.

This paper evaluated the efficiency of EU countries (Decision Making Unit, DMU). The $n$ is considered as countries ($DMU_j$, $j = 1, 2, \ldots, n$), where each consumes $m$ different inputs ($x_{ij}, i = 1, 2, \ldots, m$) and produces $s$ different outputs ($y_{rj}, r = 1, 2, \ldots, s$). The matrix of inputs may be marked as $X = \{x_{ij}, i = 1, 2, \ldots, m; j = 1, 2, \ldots, n\}$, while the matrix of outputs may be marked as $Y = \{y_{rj}, r = 1, 2, \ldots, s; j = 1, 2, \ldots, n\}$. The next aspect was the choice of returns to scale assumption. In the literature, e.g., Simar and Wilson [49], there are different ways how to decide on this assumption. They pointed to the fact that the Banker's test or bootstrap method can be used. As there was a need to normalise input and output data due to different units in our paper, we used the assumption of constant returns to scale. As mentioned by Jacobs et al. [50], the choice of constant returns to scale or variable returns to scale will usually depend on the context and purpose of the analysis, or whether short-run or long-run efficiency is under scrutiny. For example, from a societal perspective, interest may be in productivity, regardless of the scale of operations, so that constant returns to scale may be more appropriate. The second point is that a complication to the choice of constant returns to scale or variable returns to scale is that often data take the form of ratios rather than absolute numbers as measures of inputs and outputs in DEA. Also, it is the case of expenditures. Thus, when there are data in the form of ratios, it automatically implies an assumption of constant returns to scale, because the creation of the ratio removes any information about the size of the country. In this sample, there were some data in the form of ratios, but also some data in the form of absolute numbers. Therefore, to make the same format of the data, an empirical normalisation was applied. The normalised data had a comparable form as the ratios. Therefore, it may be supposed that the constant returns to scale model was more appropriate.

This assumption was combined with a non-radial and non-oriented slack-based model (SBM) model in evaluating the European Union countries between the years 2010 and 2017. In the case of the non-oriented model, the desire to improve the input side and output side at the same time was captured.

Tone [51] proposed a slack-based model (SBM) to measure efficiency in different sectors of the national economy. This model assumes that the data set is positive, i.e., $X > 0$ and $Y > 0$, and there are non-negative slacks $s_r^+$, $s_i^-$, where the slacks indicate the input excess and output shortfall. To calculate the efficiency of $DMU_q$ $(x_{iq}, y_{rq})$, the following minimisation program was used:

$$\text{Minimise} \qquad p = \frac{1 - \frac{1}{m}\sum_{i=1}^{m}(s_i^- / x_{iq})}{1 + \frac{1}{s}\sum_{r=1}^{s}(s_r^+ / y_{rq})}, \qquad (1)$$

$$\text{Subject to} \qquad \begin{array}{ll} x_{iq} = \sum_{j=1}^{n} x_{ij}\lambda_j + s_i^- & i = 1, 2, \ldots, m \\ y_{rq} = \sum_{j=1}^{n} y_{rj}\lambda_j - s_r^+ & r = 1, 2, \ldots, s \\ \lambda_j, s_r^+, s_i^- \geq 0, \end{array}$$

where $\lambda_j$ is the weight assigned to the $DMU_j$.

A country is fully SBM-efficient if $p^*$ is equal to one and all slack variables are equal to zero. This means that there is no input excess and no output shortfalls in any optimal solution. If the slack variables are not equal to zero and $p^*$ is lower than one, it means that it is necessary to make non-radial shift expressed by slack variables to achieve efficiency. For each inefficient country, we may make SBM-projection on the efficiency frontier, by eliminating the excess of inputs and increasing the output gaps.

As efficiency needed to be compared between two years, the Malmquist index was calculated to measure the productivity changes over time. The Malmquist productivity index evaluates a productivity change of a DMU between two periods as the product of "catch-up" and "frontier shift" terms. The catch-up (efficiency change) term reflects the degree that a DMU attains for improving its efficiency. In contrast, the frontier shift (innovation or technological change) term demonstrates the difference in the efficient frontier surrounding the DMU between the two periods. According to Färe and Grosskopf [52], the following formula for the computation of the Malmquist index is obtained:

$$Malmquist\ Index(x^{t+1}, y^{t+1}, x^t, y^t) = C_q \times F_q \qquad (2)$$

$$Malmquist\ Inde(x^{t+1}, y^{t+1}, x^t, y^t) = \frac{D_q^{t+1}(x^{t+1}, y^{t+1})}{D_q^t(x^t, y^t)} \times \left[\frac{D_q^t(x^{t+1}, y^{t+1})}{D_q^{t+1}(x^{t+1}, y^{t+1})} \times \frac{D_q^t(x^t, y^t)}{D_q^{t+1}(x^t, y^t)}\right]^{1/2}. \qquad (3)$$

The first component of the Malmquist index is the "catch-up" effect ($C_q$), and the second one is the "frontier-shift" effect ($F_q$). The catch-up effect explains the change over time in the efficiency of each DMU individually, while the frontier-shift effect explains the change in the best practice frontier over time, typically due to changes in technology. To fully evaluate the productivity change, we had to take into account both effects.

If the catch-up effect value is greater than one, it indicates the progress in the relative efficiency. A value equal to one indicates no changes in the relative efficiency, and a value below one indicates a regress in relative efficiency from period $t$ to period $t + 1$. A frontier-shift effect higher than one indicates progress in the frontier technology, while a value lower than one indicates regress in the frontier technology around the evaluated production unit from period $t$ to period $t + 1$. A Malmquist index higher than one indicates progress in the total factor productivity change of the evaluated production unit, while a Malmquist index equal to one shows a status quo and a Malmquist index lower than one means deterioration in the total factor productivity from period $t$ to period $t + 1$.

Within the first step, input and output variables for the efficiency model needed to be set up. As mentioned by Cook et al. [53], the selection of inputs and outputs is a very sensitive issue. It is well known that a large number of inputs and outputs compared to the number of DMUs may diminish the discriminatory power of the DEA model. The total number of inputs and outputs need to be minimised in order to increase the explanatory power of the estimated model. The reason is that with an increasing number of inputs and outputs, the number of limiting conditions that need to be expressed by the

efficiency frontier also increases. Therefore, it is recommended that the total number of inputs and outputs does not exceed 1/3 of the number of DMUs examined. The efficiency in the selected set, which comprised 28 EU countries (Belgium, Bulgaria, Czech Republic, Denmark, Germany, Estonia, Ireland, Greece, Spain, France, Croatia, Italy, Cyprus, Latvia, Lithuania, Luxembourg, Hungary, Malta, the Netherlands, Austria, Poland, Portugal, Romania, Slovenia, Slovakia, Finland, Sweden, and the United Kingdom) was examined. As the number of analysed countries was 28, it means that the number of inputs and outputs should not be higher than nine. It was decided to use no more than five indicators, as other considered indicators published within Eurostat and Scimago had the same informative value.

*2.2. Data*

Once the existing literature was studied, two models were prepared to evaluate research and development productivity in EU countries. Model 1 focused on the productivity of R&D in the public sector by using three input and two output indicators. Model 2 presents the productivity of R&D in the business (private) sector that is evaluated by two input and two output indicators. Table 1 presents the used input and output indicators of R&D for Model 1 (efficiency in the public sector) and Model 2 (efficiency in the private sector). The input and output variables were selected according to the studies mentioned in the literature review (e.g., [5,28,38,39,44–46]). There are standard indicators used in the evaluation of efficiency in the R&D area. From a quality point of view, there is always a question of which indicator is the best one to display the quality of the R&D spending incurred. According to the previous studies, it is supposed that the number of published scientific papers and citable documents, or a total number of patent applications and a high-tech export may adequately describe the quality of R&D spending. Also e.g., Dobrzanski [30] evaluated innovation expenditures efficiency in Central and Eastern European Countries by using DEA methodology.

**Table 1.** Input and output indicators used for research and development (R&D).

| | **Model 1** | **Unit** | **Source** |
|---|---|---|---|
| Input indicators | Public expenditure on R&D (government expenditure and higher education expenditure on R&D) | as % of GDP | Eurostat |
| | Total researchers in the public sector (FTE*) | Number | Eurostat |
| | Total government budget appropriations or outlays on R&D (GBAORD) | as a % of total general government expenditure | Eurostat |
| Output indicators | Scientific documents | Number | Scimago |
| | Citable documents | Number | Scimago |
| | **Model 2** | **Unit** | **Source** |
| Input indicators | R&D expenditure in the business enterprise sector (BERD) | as % of GDP | Eurostat |
| | Total researchers in the business enterprise sector (FTE*) | Number | Eurostat |
| Output indicators | Patent applications to the EPO by priority year | Number | Eurostat |
| | High-tech export | % of total export | Eurostat |

\* (FTE- Full-time equivalent) corresponds to one year's work by one person (for example, a person who devotes 40% of his time to R&D is counted as 0.4 FTE). Source: Authors according to Eurostat [54] and SJR [55].

Within the second step, data for the respective input and output variables of the EU countries were collected. Also, data from related databases and official websites were collected. Data for the years 2010, 2013, 2014, and 2017 are available at Eurostat (statistic database—science and technology) and Scopus database (Scimago Journal and Country Rank). Following the existing literature (e.g., [45,56]), we lagged the inputs by three years as compared to outputs. The reason for this can be explained by the following example. We assumed that in the first year, the researchers obtained financial resources from a grant and started to work on their research question. Also, in the second year, they prepared their

paper and sent it in to the journal. In recent years the standard processing time has been at least one year; therefore, we supposed that scientific papers will be published in the third year. Thus, the input data were relevant for years 2010 and 2014, while the data for the outputs were for 2013 and 2017, respectively. The reason for lagging the input data is that it was not possible to suppose that investment into the research and development has an effect in the form of publications or patents in the same year. Descriptive statistic of average input and output values during the whole analysed period are presented in Table 2.

**Table 2.** Descriptive statistics on input and output variables.

|  |  | Min | Max | Average | St. Dev. |
|---|---|---|---|---|---|
| Input indicators | Public expenditure on R&D as government expenditure and higher education expenditure on R&D | 0.22 | 1.10 | 0.61 | 0.23 |
|  | R&D expenditure in the business enterprise sector | 0.08 | 2.58 | 0.93 | 0.68 |
|  | Total researchers in the public sector | 247 | 171,607 | 31,589 | 43,314 |
|  | Total researchers in the business enterprise sector | 194 | 198,076 | 28,018 | 45,534 |
|  | Total GBAORD | 0.35 | 2.04 | 1.17 | 0.44 |
| Output indicators | Scientific documents | 625 | 216,739 | 39,765 | 54,105 |
|  | Citable documents | 489 | 173,978 | 34,729 | 45,849 |
|  | Patent applications to the EPO | 4.87 | 21,427.14 | 1990.92 | 4108.62 |
|  | High-tech exports | 2.70 | 34.50 | 11.84 | 6.76 |

Some conditions need to be verified before the DEA model is applied. The primary condition is that there must be a correlation between input and output variables. If the input does not affect any output, it signalises that the set of outputs is incomplete. On the other hand, if there is a high correlation between inputs (or outputs), it signalises that any of the inputs or outputs are unnecessary. Therefore, the correlation coefficient between variables during the analysed period was calculated. To find out the level of correlation, we applied the standard methodology presented by Cohen [24], who classified level of correlation according to the value of the Pearson correlation coefficient. The results can be seen in Tables 3 and 4. According to this classification, we can see moderate correlation between inputs and outputs in case of Model 1, except for input "Total researchers in the public sector", where the correlation between input and both outputs can be considered as very high. A similar situation can be seen in Model 2, where the highest correlation can be observed between "Total researchers in the business enterprise sector" and "Patent applications to the European Patent Office (EPO)"; in other cases the correlation can be considered as moderate. So, we can conclude that all our inputs affect at least one output. On the output side in Model 1, we can see a very high correlation between Scientific and Citable documents, which can signalise that one of these outputs can be considered as unnecessary. Based on the literature review dealing with this topic, we can find both these outputs as relevant for our analysis. Therefore, we decided not to exclude any of them.

As can be seen, the input and output variables were expressed in different units, and the selected countries had different sizes. In the literature, there are different ways to eliminate the problem of different sizes of countries. One of them is to use ratios instead of volume indicators, and the second one is normalisation. In this paper, it was decided to normalise the values of the indicators by using an empirical normalisation. Consequently, it was possible to compare countries with different sizes, as all input and output values were put into the interval between 0 and 1. This made it possible to compare countries with "similar size". Therefore, an empirical normalisation, which compares individual data with minimum and maximum within the dataset, was applied. As in the case of DEA, the data should

not be negative or equal to zero; the following form of empirical normalisation, which located all data within the interval from one to two, was used:

$$I_{it}^n = \frac{I_{it} - Min(I_i)}{Max(I_i) - Min(I_i)} + 1. \tag{4}$$

The next aspect was the orientation of the model. In the input-oriented model, the countries which use the minimum level of inputs to produce a given level of outputs are the primary focus. In the output-oriented model, the countries that produce the maximum level of outputs with a given level of inputs are the principal focus. In this paper, a non-oriented model was applied, as the desire to improve the input side and output side at the same time was important. The DEA was the method for relative efficiency measurement. It means that the efficiency was calculated in the specified sample and under the given inputs and outputs. Every change in the dataset will lead to different results.

**Table 3.** Correlation analysis between input and output variables in the public sector.

| | Public expenditure on R&D as government expenditure and higher education expenditure on R&D | Total researchers in the public sector | Total GBAORD | Scientific documents | Citable documents |
|---|---|---|---|---|---|
| | **Public Sector—Model 1** | | | | |
| Public expenditure on R&D as government expenditure and higher education expenditure on R&D | 1 | | | | |
| Total researchers in the public sector | 0.235545 | 1 | | | |
| Total GBAORD | 0.833645 | 0.303632 | 1 | | |
| Scientific documents | 0.288376 | 0.979799 | 0.349403 | 1 | |
| Citable documents | 0.302302 | 0.977897 | 0.362126 | 0.997663 | 1 |

**Table 4.** Correlation analysis between input and output variables in the private sector.

| | R&D expenditure in the business enterprise sector | Total researchers in the business enterprise sector | Patent applications to the EPO | High-tech exports |
|---|---|---|---|---|
| | **Private Business Enterprise Sector—Model 2** | | | |
| R&D expenditure in the business enterprise sector | 1 | | | |
| Total researchers in the business enterprise sector | 0.437253 | 1 | | |
| Patent applications to the EPO | 0.430536 | 0.948173 | 1 | |
| High-tech exports | 0.138135 | 0.239096 | 0.20138 | 1 |

## 3. Results

An empirical analysis focused on R&D efficiency in the public and private sectors of EU countries (28) by using DEA methodology during 2010/2013 and 2014/2017. The R&D efficiency in EU countries was evaluated by selected input and output indicators and changes of R&D efficiency were evaluated based on two models (in public and private sectors) by using the Malmquist index during 2010/2013 and 2014/2017. The efficiency score, Malmquist index, and its components were calculated by mathematical program DEA Solver Pro 13.

### 3.1. An Evaluation of R&D Efficiency in the Public Sector of EU Countries

Model 1 provided an evaluation of R&D efficiency in the public sector, where a time delay in outputs was present. Science and research results or number of researches generate certain publications and citable publications that are mostly published with a three-year delay. The inputs were three years older, i.e., the inputs used for the year 2014 and outputs used for the year 2017 for the second observation year in order to use available data of publications and citable publications in 2017. Table 5 presents the results of R&D efficiency in the public sector based on selected input indicators

(I1: Public R&D expenditure in the Government sector and in the Higher education sector as % GDP, I2: Total researchers in the public sector (FTE), I3: Total GBAORD as a % of total general government expenditure) and output indicators (O1: Scientific documents and O2: citable documents from Scopus database) in EU countries during 2010/2013 and 2014/2017 by using DEA methodology. The efficiency of 2010/2013 means that the input values were used from 2010 and the output values were used from 2013. The second observation year was 2014 (inputs) and 2017 (outputs).

**Table 5.** The R&D efficiency scores in the public sector of EU countries.

| No. | DMU | 2010/2013 | Rank | 2014/2017 | Rank |
|-----|-----|-----------|------|-----------|------|
| 1 | Belgium | 0.7473 | 15 | 0.7216 | 16 |
| 2 | Bulgaria | 0.7669 | 12 | 0.786 | 8 |
| 3 | Czechia | 0.7157 | 18 | 0.6649 | 23 |
| 4 | Denmark | 0.6579 | 26 | 0.6363 | 26 |
| 5 | Germany | 0.8838 | 3 | 0.8215 | 6 |
| 6 | Estonia | 0.6229 | 27 | 0.6059 | 28 |
| 7 | Ireland | 0.7695 | 10 | 0.7278 | 15 |
| 8 | Greece | 0.804 | 6 | 0.7189 | 18 |
| 9 | Spain | 0.7629 | 13 | 0.8329 | 5 |
| 10 | France | 0.8756 | 4 | 0.855 | 3 |
| 11 | Croatia | 0.6971 | 21 | 0.6927 | 19 |
| 12 | Italy | 1 | 1 | 1 | 1 |
| 13 | Cyprus | 0.7454 | 16 | 0.7667 | 11 |
| 14 | Latvia | 0.7908 | 8 | 0.7749 | 10 |
| 15 | Lithuania | 0.6871 | 23 | 0.6647 | 24 |
| 16 | Luxembourg | 0.6841 | 24 | 0.6441 | 25 |
| 17 | Hungary | 0.7527 | 14 | 0.7893 | 7 |
| 18 | Malta | 0.8008 | 7 | 0.7784 | 9 |
| 19 | The Netherlands | 0.7793 | 9 | 0.7547 | 13 |
| 20 | Austria | 0.6881 | 22 | 0.6695 | 22 |
| 21 | Poland | 0.7683 | 11 | 0.7618 | 12 |
| 22 | Portugal | 0.699 | 20 | 0.7387 | 14 |
| 23 | Romania | 0.8112 | 5 | 0.841 | 4 |
| 24 | Slovenia | 0.6772 | 25 | 0.7193 | 17 |
| 25 | Slovakia | 0.7382 | 17 | 0.683 | 21 |
| 26 | Finland | 0.6016 | 28 | 0.6245 | 27 |
| 27 | Sweden | 0.7015 | 19 | 0.6879 | 20 |
| 28 | The United Kingdom | 1 | 1 | 1 | 1 |
| | Average | 0.7582 | | 0.7486 | |

Source: Authors' calculation according to Eurostat [54] and SJR [55].

Table 5 shows that the average efficiency score reached 75.82% in the first analysed year. However, 13 EU countries achieved efficiency score higher than average value, and in the case of 15 EU countries, efficiency score was lower. There were only two countries that reached efficiency score equal to value 1 in a given year: Italy and the United Kingdom. Finland achieved the lowest efficiency score. It was probably influenced by inputs, which reached the maximum values from all of the analysed countries (RD exp 1.1%, GBOARD 2.04%), while in case of outputs, Finland belonged to a group of countries with a below-average number of publications and citable publications.

In the second analysed year, an average efficiency score reached 74.86%. Even this year, there were only two countries that reached efficiency score equal to value 1 (Italy and the United Kingdom). Consequently, they may be considered as efficient in a transformation of their inputs to outputs. Also, this year, 13 countries achieved a higher score than average and 15 countries with a lower score than average value. However, Finland belonged to the countries with the lowest efficiency score. In the second analysed year, Estonia had the lowest efficiency score due to low values of outputs in comparison to very high input values within the analysed group of countries. Table 6 provides

an overview of evaluated changes in R&D efficiency in the public sector by using the Malmquist index during 2010/2013 and 2014/2017.

**Table 6.** Changes of R&D efficiency by using the Malmquist index in the public sector during 2010/2013 and 2014/2017.

| No. | DMU | Catch-Up | Frontier-Shift | MI | Rank |
|-----|-----|----------|----------------|-----|------|
| 1 | Belgium | 0.9656 | 1.005 | 0.9704 | 20 |
| 2 | Bulgaria | 1.0249 | 1.0046 | 1.0296 | 8 |
| 3 | Czechia | 0.929 | 1.0044 | 0.9331 | 25 |
| 4 | Denmark | 0.9671 | 1.0048 | 0.9717 | 19 |
| 5 | Germany | 0.9295 | 1.0054 | 0.9345 | 24 |
| 6 | Estonia | 0.9727 | 1.0038 | 0.9765 | 17 |
| 7 | Ireland | 0.9458 | 1.0052 | 0.9508 | 22 |
| 8 | Greece | 0.8942 | 1.0037 | 0.8974 | 27 |
| 9 | Spain | 1.0919 | 1.0052 | 1.0975 | 1 |
| 10 | France | 0.9764 | 1.0057 | 0.982 | 15 |
| 11 | Croatia | 0.9937 | 1.0054 | 0.9991 | 10 |
| 12 | Italy | 1 | 0.9979 | 0.9979 | 11 |
| 13 | Cyprus | 1.0286 | 1.0052 | 1.0339 | 7 |
| 14 | Latvia | 0.9799 | 1.0026 | 0.9824 | 14 |
| 15 | Lithuania | 0.9674 | 1.003 | 0.9703 | 21 |
| 16 | Luxembourg | 0.9415 | 1.0043 | 0.9455 | 23 |
| 17 | Hungary | 1.0485 | 1.0035 | 1.0522 | 4 |
| 18 | Malta | 0.9721 | 1.0041 | 0.9761 | 18 |
| 19 | Netherlands | 0.9685 | 0.8835 | 0.8557 | 28 |
| 20 | Austria | 0.973 | 1.0052 | 0.9781 | 16 |
| 21 | Poland | 0.9914 | 1.0039 | 0.9953 | 12 |
| 22 | Portugal | 1.0568 | 1.0026 | 1.0596 | 3 |
| 23 | Romania | 1.0368 | 1.005 | 1.042 | 5 |
| 24 | Slovenia | 1.0621 | 1.0036 | 1.066 | 2 |
| 25 | Slovakia | 0.9252 | 1.004 | 0.9289 | 26 |
| 26 | Finland | 1.0381 | 1.0032 | 1.0414 | 6 |
| 27 | Sweden | 0.9806 | 1.0081 | 0.9886 | 13 |
| 28 | United Kingdom | 1 | 1.0023 | 1.0023 | 9 |
| | Average | 0.9879 | 0.9998 | 0.9878 | |

The results of the Malmquist index indicate that the total productivity decreased by 1.22% during the two analysed periods. This process started with a decrease in technical efficiency by 1.21% as well as due to a decrease in innovation activity by 0.02% (Table 6). It may be stated that in the case of nine countries, a growth of total productivity, which was measured by the Malmquist index, was observed. Here belong the following countries: Bulgaria, Spain, Cyprus, Hungary, Portugal, Romania, Slovenia, Finland, and the United Kingdom. On the other hand, there are 19 countries where a decrease in total productivity measured by the Malmquist index was observed.

The highest value of progress between two analysed periods was evident in Spain by means of catch-up as well as frontier-shift effects. In the case of Spain, progress of 9.75% was recognised, while a technical efficiency grew of 9.19% and technological progress grew by 0.52% as a consequence of shifting the efficiency frontier. This positive development was determined by a significant decrease at the inputs' side (decrease I1 by 10.77%, I2 by 13.07%, and I3 by 26.19%) that was connected with an increase on the outputs' side (increase O1 of 8.65% and O2 of 8.18%). On the other hand, a decrease was evident in the case of the Netherlands of 14.43%, that was caused by a decrease in technical efficiency of 3.15% and also by a decrease in innovation activity of 11.65%. This negative trend began due to growth of one of the inputs (increase I2 of 12.65%) and without any reduction in the case of the third input that indicated a slight increase on the outputs' side (6.75%, or 6.32%).

The catch-up effect and the frontier-shift effect describe a mutual relationship between the components of the Malmquist index according to R&D efficiency in the public sector of EU countries. Figure 1 presents this relationship. The frontier-shift effect represents an improvement in efficiency due to innovation (only in the case of Italy and the Netherlands was a decrease evident). On the contrary, catch-up effect represents an improvement in efficiency due to improved operations and management of public sector (it may be seen in the case of Bulgaria, Spain, Cyprus, Hungary, Portugal, Romania, Slovenia, Finland). Figure 1 may be divided into four quadrants based on growth or a decline in technical efficiency of technological progress. Within the first quadrant, there are countries which obtained progress in efficiency due to the innovation (frontier shift effect is higher than one) and a decrease in efficiency due to worsening operations and management of public sector (catch-up effect is lower than one). Within the first quadrant, there are countries like Belgium, Czech Republic, Denmark, Germany, Estonia, Ireland, Greece, France, Croatia, Latvia, Lithuania, Luxembourg, Malta, Austria, Poland, Slovakia, and Sweden. The second quadrant represents countries which obtained progress in efficiency due to innovation (frontier shift effect is higher than one) and also progress in efficiency due to improved operations and management of the public sector (catch-up effect is higher than one). Within the second quadrant, there are countries like Bulgaria, Spain, Cyprus, Hungary, Portugal, Romania, Slovenia, Finland, and the United Kingdom. The third quadrant represents countries which obtained a decrease in efficiency due to the innovation (frontier shift effect is lower than one) and also a decrease in efficiency due to worsening operations and management of public sector (catch-up effect is lower than one), where only the Netherlands is located. The last quadrant represents countries which obtained a decrease in efficiency due to the innovation (frontier shift effect is lower than one), while they obtained progress in efficiency due to improved operations and management of public sector (catch-up effect is higher than one), where only Italy may be seen.

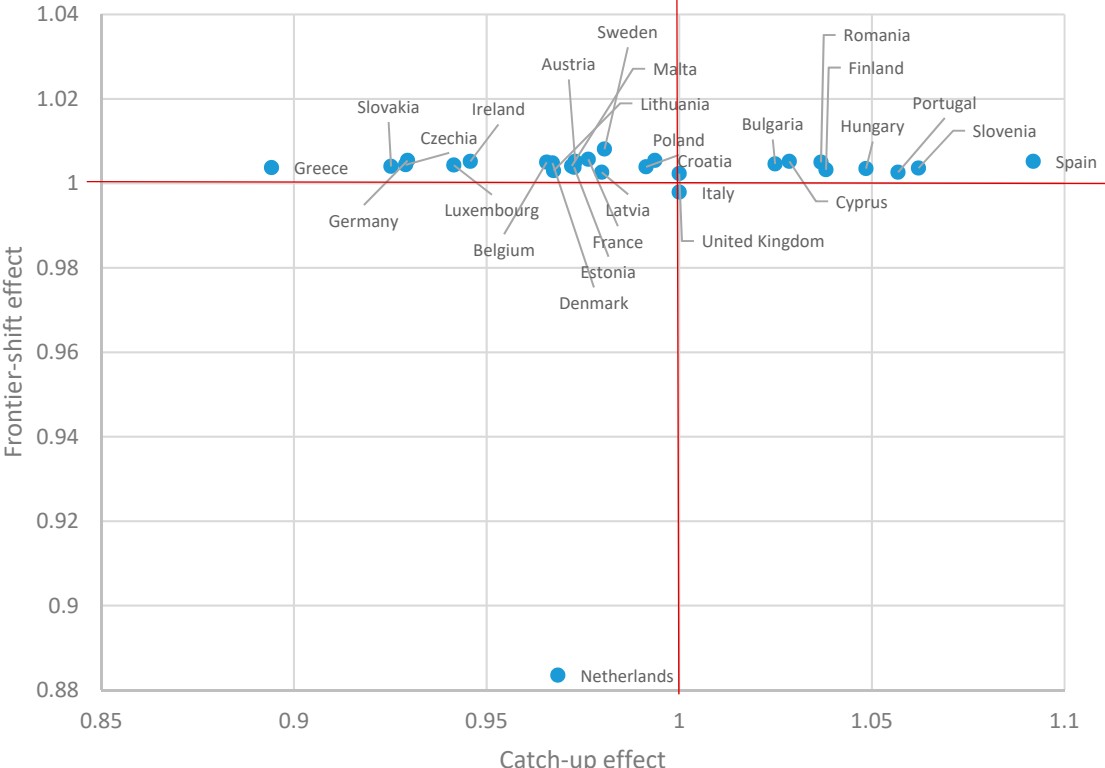

**Figure 1.** Changes of R&D efficiency in the public sector of EU countries—relationship between the components of the Malmquist index during 2010/2013 and 2014/2017.

### 3.2. An Evaluation of R&D Efficiency in Private Sector of EU Countries

This part presents an overview of the evaluation of research and development efficiency in the private sector. Also, a certain time delay was considered in this case. This research was based on the fact that all science and research costs, and/or a number of researchers generate certain patents and high-tech exports, are most evident with a three-year delay. Evaluation of R&D efficiency in the private sector used DEA methodology and normalised values, which were empirically investigated; model of constant returns to scale. The SBM non-oriented model was selected. This model proposes changes on both sides of inputs and outputs at the same time in order to achieve efficiency.

Table 7 presents the results of R&D efficiency score in the private sector of EU countries, including the rank of countries during 2010/2013 and 2014/2017. The entry Efficiency 2010/2013 provides input values that were used from 2010 and output values from 2013. The next observation year was 2014 (inputs) and 2017 (outputs).

**Table 7.** R&D efficiency score in the private sector of EU countries.

| No. | DMU | 2010/2013 | Rank | 2014/2017 | Rank |
|-----|-----|-----------|------|-----------|------|
| 1 | Belgium | 0.641 | 22 | 0.649 | 24 |
| 2 | Bulgaria | 0.7137 | 18 | 0.7317 | 19 |
| 3 | Czechia | 0.7754 | 12 | 0.7656 | 15 |
| 4 | Denmark | 0.6037 | 27 | 0.6353 | 25 |
| 5 | Germany | 1 | 1 | 1 | 1 |
| 6 | Estonia | 0.7909 | 11 | 0.81 | 11 |
| 7 | Ireland | 0.8137 | 6 | 1 | 1 |
| 8 | Greece | 0.7061 | 19 | 0.7953 | 12 |
| 9 | Spain | 0.619 | 25 | 0.6736 | 22 |
| 10 | France | 0.7052 | 20 | 0.7295 | 20 |
| 11 | Croatia | 0.7624 | 15 | 0.8165 | 10 |
| 12 | Italy | 0.7231 | 16 | 0.7606 | 16 |
| 13 | Cyprus | 1 | 1 | 1 | 1 |
| 14 | Latvia | 0.8135 | 7 | 0.9158 | 6 |
| 15 | Lithuania | 0.7652 | 14 | 0.8302 | 8 |
| 16 | Luxembourg | 0.8442 | 5 | 0.7375 | 18 |
| 17 | Hungary | 0.8041 | 9 | 0.7815 | 13 |
| 18 | Malta | 1 | 1 | 1 | 1 |
| 19 | The Netherlands | 1 | 1 | 0.8206 | 9 |
| 20 | Austria | 0.6702 | 21 | 0.6682 | 23 |
| 21 | Poland | 0.7944 | 10 | 0.7699 | 14 |
| 22 | Portugal | 0.6252 | 24 | 0.6957 | 21 |
| 23 | Romania | 0.7729 | 13 | 0.9262 | 5 |
| 24 | Slovenia | 0.6132 | 26 | 0.6097 | 27 |
| 25 | Slovakia | 0.8121 | 8 | 0.8651 | 7 |
| 26 | Finland | 0.5397 | 28 | 0.5883 | 28 |
| 27 | Sweden | 0.6384 | 23 | 0.6282 | 26 |
| 28 | The United Kingdom | 0.7139 | 17 | 0.7397 | 17 |
| | Average (EU28) | 0.7593 | | 0.7837 | |

Source: Authors' calculation according to Eurostat [54].

As Table 7 shows, the average efficiency score reached a value of 75.93%. Fifteen countries reached an efficiency score higher than the average value and 13 countries reached lower efficiency scores. In a selected year, four countries achieved efficiency score equal to value 1: Germany, Cyprus, Malta and the Netherlands. Finland reached the lowest efficiency score. This might have been influenced by science and research costs that reached the maximum values from all of the analysed countries (RD exp. 2.5%). However, Finland belonged to the group of countries with a below-average number of patents and high-tech export on the outputs' side. In the second analysed year, the average efficiency

score reached a value of 78.37%. There were four countries (Germany, Ireland, Cyprus, and Malta) that reached efficiency score equal to value 1. Consequently, these countries may be considered efficient in the transformation of their inputs to outputs. In this particular year, 12 countries achieved a higher score than average and 16 countries reached a lower score than average value. Finland belonged to the countries with the lowest efficiency score. Subsequently, the Malmquist index was calculated, which reflected changes in R&D efficiency in the private sector of EU countries during 2010/2013 and 2014/2017. It is presented in Table 8.

**Table 8.** Changes of R&D efficiency by using the Malmquist index in the private sector during 2010/2013 and 2014/2017.

| No. | DMU | Catch-Up | Frontier-Shift | MI | Rank |
|-----|-----|----------|----------------|------|------|
| 1 | Belgium | 1.0124 | 0.8979 | 0.9091 | 15 |
| 2 | Bulgaria | 1.0253 | 0.883 | 0.9053 | 16 |
| 3 | Czechia | 0.9873 | 0.9057 | 0.8942 | 21 |
| 4 | Denmark | 1.0524 | 0.8988 | 0.9459 | 10 |
| 5 | Germany | 1 | 0.8164 | 0.8164 | 26 |
| 6 | Estonia | 1.0241 | 0.9042 | 0.926 | 12 |
| 7 | Ireland | 1.2289 | 0.8874 | 1.0906 | 1 |
| 8 | Greece | 1.1262 | 0.8653 | 0.9745 | 5 |
| 9 | Spain | 1.0881 | 0.8929 | 0.9715 | 6 |
| 10 | France | 1.0344 | 0.8739 | 0.904 | 17 |
| 11 | Croatia | 1.0709 | 0.8879 | 0.9508 | 7 |
| 12 | Italy | 1.0518 | 0.8572 | 0.9017 | 19 |
| 13 | Cyprus | 1 | 0.9016 | 0.9016 | 18 |
| 14 | Latvia | 1.1258 | 0.8426 | 0.9486 | 8 |
| 15 | Lithuania | 1.085 | 0.8737 | 0.948 | 9 |
| 16 | Luxembourg | 0.8736 | 0.8666 | 0.7571 | 28 |
| 17 | Hungary | 0.9719 | 0.9063 | 0.8809 | 24 |
| 18 | Malta | 1 | 0.915 | 0.915 | 14 |
| 19 | The Netherlands | 0.8206 | 0.9471 | 0.7772 | 27 |
| 20 | Austria | 0.997 | 0.9028 | 0.9001 | 20 |
| 21 | Poland | 0.9691 | 0.8704 | 0.8435 | 25 |
| 22 | Portugal | 1.1128 | 0.8938 | 0.9947 | 3 |
| 23 | Romania | 1.1984 | 0.8598 | 1.0303 | 2 |
| 24 | Slovenia | 0.9943 | 0.8968 | 0.8917 | 22 |
| 25 | Slovakia | 1.0652 | 0.8795 | 0.9369 | 11 |
| 26 | Finland | 1.09 | 0.8958 | 0.9764 | 4 |
| 27 | Sweden | 0.9841 | 0.8992 | 0.8849 | 23 |
| 28 | The United Kingdom | 1.0361 | 0.8884 | 0.9204 | 13 |
| | Average (EU28) | 1.0366 | 0.8861 | 0.9178 | |

Table 8 presents the Malmquist index results that the total productivity of research and development decreased by 8.22% during two analysed periods. This decrease was caused by a decrease in innovation activity by 11.39% and slightly positively influenced by a technical efficiency growth of 3.66%. It may be stated that a growth of total productivity measured by the Malmquist index was observed only in two countries: Ireland and Romania. The rest of the countries showed a decrease in the total productivity of research and development measured by the Malmquist index. The highest progress value was observed in Ireland by means of a high catch-up effect between two analysed periods. In the case of Ireland, it increased by 9.06%, and a technical efficiency also increased by 22.89%. However, it was reduced by a decrease due to shifting efficiency frontier by means of technological progress of 11.26%. This positive development was evident in a significant growth on the outputs' side (an increase of O1 of 12.86%, O2 of 65.07%). On the other hand, an evident decrease between the analysed periods was obvious in Luxembourg (24.29%). A technical efficiency decrease caused this decrease by 12.64%

and innovation activity decrease by 13.34%. The negative trend was influenced by a decrease in the outputs' side, where the number of patents decreased by 14.63% and HT export value by 68.49%.

The overall Malmquist index (MI) may be divided into two effects: frontier-shift effect and catch-up effect. The frontier-shift effect represents an improvement in efficiency due to the innovation (in the case of all countries, a decrease was evident). On the contrary, catch-up effect represents an improvement in efficiency due to improved operations and management of private sector (a decrease may be seen in the case of eight countries: Czech Republic, Luxembourg, Hungary, the Netherlands, Austria, Poland, Slovenia and Sweden; in the case of other countries progress may be seen).

Also, Figure 2 may be divided into four quadrants based on growth or a decline in technical efficiency of technological progress. Within the first quadrant, there are countries which reflect progress in efficiency due to the innovation and decrease in efficiency due to worsening operations and management of the public sector. However, in this sample, there is no country within this quadrant. The second quadrant represents countries which obtained progress in efficiency due to innovation and also progress in efficiency due to improved operations and management of the public sector. However, there is no country even within this quadrant. The third quadrant represents countries which obtained a decrease in efficiency due to the innovation and also a decrease in efficiency due to worsening operations and management of the public sector. There are countries like the Czech Republic, Luxembourg, Hungary, the Netherlands, Austria, Poland, Slovenia, and Sweden. The last quadrant represents countries which obtained a decrease in efficiency due to the innovation, while they obtained progress in efficiency due to improved operations and management of the public sector. Within the fourth quadrant, there are countries like Belgium, Bulgaria, Germany, Estonia, Ireland, Greece, Spain, France, Croatia, Italy, Cyprus, Latvia, Lithuania, Malta, Portugal, Romania, Slovakia, Finland, and the United Kingdom.

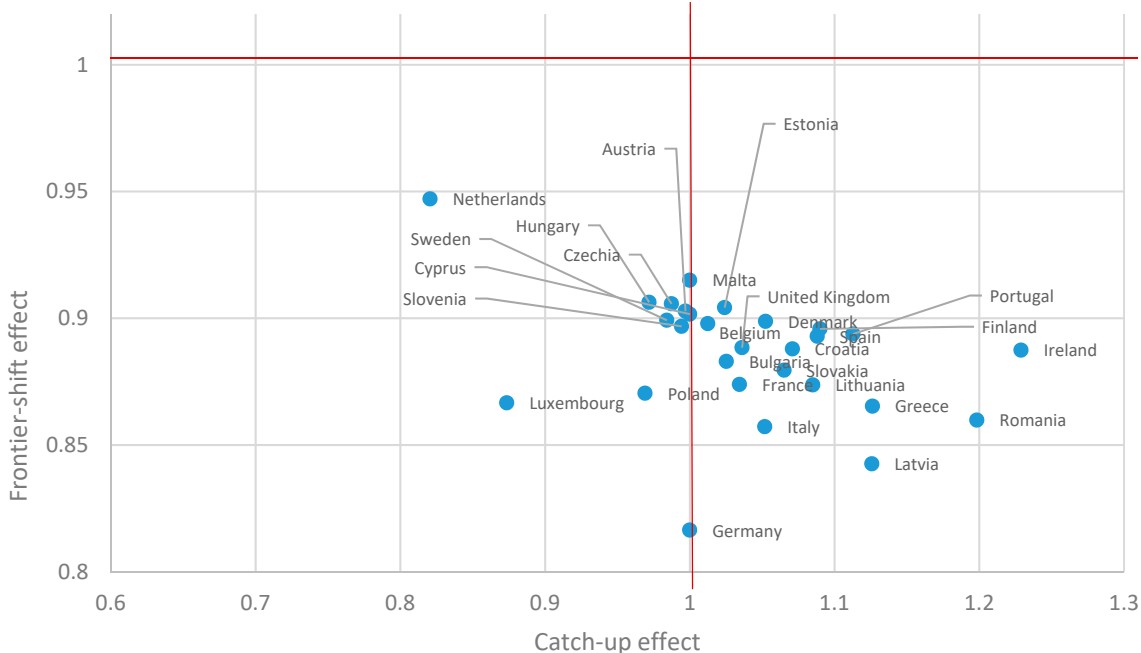

**Figure 2.** Changes of R&D efficiency in the private sector of EU countries—Relationship between the components of the Malmquist index during 2010/2013 and 2014/2017.

## 4. Discussion

At present, the highest priority in the research and development sector, in the European dimension, is an interconnection of obtained scientific knowledge with its subsequent use in practice [2]. Many researches have dealt with research and development efficiency in the individual countries at the national or regional level. However, less attention has been paid to evaluating the efficiency of R&D in

the public and the private sector. As some authors suggested, e.g., [50], the importance of public vs. private R&D is country-specific and should, therefore, be taken into account when measuring research efficiency. These authors distinguish between R&D expenditures conducted by business enterprises, by the government, and by the higher-education sector and other indicators.

This research examined three research questions. For the research question (RQ1) "Are the European countries efficient in the process of transformation of investment into the research and development in the outputs in the form of scientific and citable documents or patens and high-tech export?" The answer is: NO. There were only two countries (Italy and the United Kingdom) out of 28 European countries in the public sector that efficiently transformed selected inputs (public expenditure R&D, researchers, or GBAORD) in 2010 and 2014 to outputs in the form of publications and citable publications in 2013 and 2017. In the case of 11 countries, in 2010/2013 and 2014/2017, an above-average R&D efficiency was determined in comparison to the countries', the EU 28s', average, i.e., 2010/2013 (75.82%) and 2014/2017 (74.86%). It may be stated that in a majority of countries, there were realised selected priorities of R&D national policies and their effort was to fulfil a trend of the Europe 2020 strategy in research and development that is connected with increasing expenditures on the public sector (e.g., Greece, Malta, Czech Republic, Slovakia, Lithuania). Rank 3 of R&D efficiency in 2010/2013 was reached by Germany and in 2014/2017 by France when considering the rank of countries according to R&D efficiency. In the public sector, in 2010/2013 and 2014/2017, Italy and United Kingdom reached Rank 1. Both Germany and France are economically developed countries with a high potential to develop scientific and research area in the public sector. This is also reflected in R&D results in the form of publications in Scopus database, but also in their quality (citable publications). The lowest R&D efficiency in the public sector was reached by Finland in 2010/2013 (Rank 28) and Estonia (Rank 27), and in 2014/2017, Estonia reached Rank 28 and Finland Rank 27. Such low R&D efficiency may be explained by their relatively high input potential of R&D in the public sector, i.e., high R&D expenditures, state expenditures, R&D subsidies (GBAORD). This potential was not effectively produced to a required number of outputs (scientific documents and citable documents).

The results of R&D efficiency in the private business enterprise sector (Table 8) show that 4 countries out of 28 in 2010 and 4 countries in 2014 efficiently transformed selected inputs (expenditure, researchers) to outputs (patents applications to the EPO and high-tech exports) in 2013 and 2017. Above-average R&D efficiency was determined in 11 countries during 2010/2013 in comparison to the EU 28 average (75.93%), apart from those four efficient countries, and six countries reached above-average R&D efficiency during 2014/2017 in comparison to the EU 28 average (78.37%). Many countries had increased their input potentials within R&D expenditures in the private business enterprise sector (similarly as in the public sector) that was a part of Strategy 2020. However, these potentials were not efficiently transformed into outputs (patents applications to the EPO and high-tech exports). Finland (Rank 28) and Slovenia (Rank 27, 26) achieved the lowest R&D efficiency in private business enterprise sector out of all EU 28 during both analysed periods. In 2010/2013, Denmark (Rank 27) reached the lowest R&D efficiency, and in 2014/2017, it was Sweden (Rank 26). It is usually caused by high R&D expenditures to private business enterprise sector that were not efficiently produced into the number of R&D results with all other input potentials (patents applications to the EPO and high-tech exports).

Considerable differences in R&D efficiency that is influenced by many factors are evident from the results. Science and research expenditures, and also R&D expenditures in the individual sectors play an important role. According to Cullmann, Schmidt-Ehmcke, and Zloczysti [57], differentiation of R&D in the public and private sector provides a more detailed picture compared to the conventional use of aggregate R&D because the distribution of R&D expenditures over sources varies remarkably across countries. The importance of public vs. private R&D is country-specific and should, therefore, be taken into account when measuring research efficiency. Furthermore, the productivity of R&D may vary across sectors—a dollar invested in private R&D might increase a country's patent output more than a dollar invested in public R&D (see Wang, 2007). The distinction between private and public R&D is especially useful since the question of whether these are complements or substitutes has not yet been

satisfactorily answered in the literature [14]. As other authors suggest (e.g., [24,25,29,58]), the scientific policy of the individual countries affects the target of research and development that is publicly funded. Thus, research in the government and the higher-education sector focuses on obtaining unique knowledge in unknown areas that contribute to knowledge growth and strengthening of innovation efficiency of companies, and also sustainable resource conservation. As the comparative analyses and researches (e.g., [2,14,59–62]) presented without necessary financial resources, both from the government and the business enterprise sector, it may not be expected that R&D will bring knowledge, innovations, and technologies competitive on an international level, which will increase productivity, employment rate, and economic competitiveness. According to Hu, Yang, and Chen [26], intellectual property rights protection, technological co-operation among business sectors, knowledge transfer between business sectors and higher-education institutions, agglomeration of R&D facilities, and involvement of the government sector in R&D activities significantly improve national R&D efficiency.

Research question (RQ2) was verifying the following: Is R&D efficiency in the public sector in the European countries during 2010/2013 and 2014/2017 influenced by technological progress? Yes (in a majority of countries). As the results in Table 8 show, a total growth of R&D efficiency in the public sector was determined in nine countries during 2010/2013 and 2014/2017. This growth was especially influenced by increasing technical efficiency, an increase of technological progress, and development of innovation activity measured by the Malmquist index. The total decline of R&D efficiency was determined in 19 countries during 2010/2014 and 2014/2017.

Most of these countries were influenced by the growth of technological progress and innovation activities during individually analysed periods. However, a technical efficiency decrease represents their main obstruction. An increase of technological progress and innovation activities in 26 countries influenced the changes in R&D efficiency, and in the case of 8 countries, these changes were accompanied by an increase in technical efficiency. Spain achieved the best rank (Rank 1), where total productivity increased by 9.75%, Slovenia (Rank 2—an increase of 6.60%), and Portugal (Rank 3—an increase of 5.96%) based on the results of evaluated changes of R&D efficiency that were measured by the Malmquist index during 2010/2013 and 2014/2017. Positive progress, in case of these countries, was caused by a decrease in an input potential and an increase in outputs and/or R&D results (scientific publications and citable document). The highest decline of R&D efficiency in the public sector during the monitored periods, 2010/2013 and 2014/2017, was determined in the Netherlands (Rank 28) and Greece (Rank 27). In the case of the Netherlands, this decline in efficiency was caused by a decrease in technological progress and innovation activities, but also by a decrease in technical efficiency in R&D. Similarly, in the case of Greece, R&D efficiency decline was a cause of decreased technical efficiency (catch-up effect). R&D efficiency decline in both countries may be partially explained by the growth of an input potential into R&D in the public sector (the Netherlands—researchers of 12.65% and Greece—expenditure of 54.29%, researchers of 20.58%, GBAORD of 50%). However, increasing the input potential was not efficiently transformed into the required number of R&D results. Consequently, it may be stated that RQ2 was confirmed in a majority of countries.

The third research question (RQ3) was verifying the following: Were the changes in R&D efficiency in private sector in the European countries significantly influenced by a technical efficiency during 2010/2013 and 2014/2017? The answer to RQ3 is YES (in a majority of countries). The growth of R&D efficiency during 2010/2013 and 2014/2017 was evident in 2 countries out of 28 based on the changes of R&D efficiency that were measured by the Malmquist index in private business enterprise sector (Table 8). Ireland (Rank 1) showed an increase of total productivity of 9.06% and Romania (Rank 2) demonstrated an increase of R&D productivity of 3.03%, which was caused by an increase of technical efficiency (catch-up effect). Positive progress that is connected with R&D efficiency increase, in the case of Romania, may be explained by inputs' decrease (expenditure by 11.1% and researchers by 10.4%) and outputs' increase (patents to applications to the EPO of 12.86% and high-tech exports of 65%). In the case of Ireland, R&D efficiency growth was caused by an increase of produced outputs in the form of patents to applications to the EPO of 12.8% and high-tech exports of 65%. The rest of

the 26 examined countries showed R&D efficiency decline in the private sector during 2010/2013 and 2014/2017 due to a decrease in technological progress and innovation activities (frontier-shift), and in the case of 8 countries, also due to a decrease in technical efficiency. However, the growth of technical efficiency caused changes in total R&D productivity in 17 examined countries. The highest decline of R&D efficiency was determined in the private sector of Luxembourg (24.29%), that was caused by a decrease in technical efficiency by 12.64% and innovation activities by 13.34%. Even if Luxembourg decreased its input potential into R&D, it also significantly decreased the outputs (patents to the EPO by 14.6% and high-tech export by 68.5%) that negatively influenced the total R&D efficiency.

The analysis results of R&D efficiency in EU countries indicate different trends in the area of R&D in the private and public sectors. There are many other authors ([23,58,63]) who evaluated R&D in the public sector (higher education and government sectors) and private sector in the European context by using similar input and output indicators as were used in this research. Also, Aristovnik [64] and Hu, Yang, and Chen [28] (as in our study) used expenditure on R&D and the number of full-time researchers as input indicators when assessing the efficiency of R&D by DEA method, and the number of scientific publications indexed in the Science Citation Index as output indicators.

Results of Conte et al. [63] indicate large cross-country differences in terms of measured efficiency. Some authors, such as Skrinjaric [23], confirmed different results in measuring efficiency when choosing different indicators and units. The efficiency results in the individual countries depend on an evaluation of an indicator, input or output, or on a model that was used (input-oriented, output-oriented, or non-oriented).

## 5. Conclusions

The debate about Industry 4.0 and its global impact is driven by uncertainty about the best way to exploit the fast pace of technological innovation to improve various aspects of human life [8,9]. The authors' effort was an evaluation of one of the Industry 4.0 areas within technology and innovation activities. R&D efficiency with an emphasis on a separate role of the public and private sectors in the EU 28 was evaluated in more detail by using the DEA method. Italy and the United Kingdom demonstrated R&D efficiency in the public sector during 2010/2013 and 2014/2017. R&D efficiency in the private business enterprise sector during 2010/2013 was determined in Germany, Cyprus, Malta, the Netherlands, and during 2014/2017 in Germany, Cyprus, Malta, and Ireland. The results show a significant decline in the total R&D productivity in EU 28 in public and private sector during 2010/2013 and 2014/2017. Even if 9 countries confirmed a total growth of R&D efficiency in the public sector, in the case of 19 countries, a total decline of R&D efficiency was evident during the monitored periods, 2010/2013 and 2014/2017. In the private sector, growth of R&D efficiency was determined during 2010/2013 and 2014/2017 in two countries. The research results show that the highest growth of total R&D productivity in the public sector was present in Spain, Slovenia, and Portugal, based on the evaluation of R&D efficiency changes during 2010/2013 and 2014/2017 by using the Malmquist index. The highest growth of R&D productivity in the private sector was determined in Ireland and Romania. These changes in R&D productivity were caused by an increase in technical efficiency (catch-up effect). The recommendations on how to increase R&D efficiency may be different in individual countries. However, most of the inefficient countries pay attention, in increasing or decreasing inputs, to increase of outputs' production, i.e., in this case, in the public sector, attention is paid to a number of scientific publications and citable documents, and in the private sector, to patents applications to the EPO and high-tech exports as % of total export. In a majority of countries, the next recommendation would be the growth of R&D efficiency and efficient use of input indicators, especially the researchers by sector performance, who are output carriers. Ultimately, EU countries' innovation policies aim to explicitly link science, technology, and innovation with economic and employment growth. Individual countries' innovation strategies must also coordinate disparate policies toward scientific research, technology commercialisation, information technology, investments, education, and skills development that drives economic growth. The future research will focus on efficiency evaluation of scientific

and research, technological, and innovation activities, so on wider connections and present issues of the Industry 4.0 (social, educational, or environmental) following the discussion topic, Industry 4.0, and its global impact. The study results provide a valuable platform for the creators of relevant policies. These results are particularly important for the creators of the national and regional strategic and innovation, investment, and educational plans. Also, the study results support the creation of strategic documents that are necessary for efficient management of the R&D system in the individual countries. This process is directly linked to the availability of a high-quality database that would enable an evaluation of measures taken and the creation of proper decision-making mechanisms. Simultaneously, this study is an appeal to the creation of an international comparative platform that would provide consistent and compatible data of R&D in the individual countries. Consequently, new possibilities that would increase the efficiency of research and development, which is required for countries' economies, may be revealed.

**Author Contributions:** Conceptualization, M.H. and B.G.; methodology, M.H.; software, K.K.; validation, B.G.; formal analysis, M.H. and B.G.; investigation, B.G.; resources, K.K.; data curation, M.H.; writing—original draft preparation, M.H. and B.G.; writing—review and editing, B.G. and M.H.; visualisation, K.K.; supervision, M.H.; project administration, B.G.; funding acquisition, B.G. All authors have read and agreed to the published version of the manuscript.

**Funding:** The research was supported by the Research and Development Agency GA AA under the contract No. 21/2020: *"Management, business risk and the firm bankruptcy in the segment of SMEs"* and as part of the research project VEGA 1/0794/18 "*Development of methodological platform for evaluation of efficiency in the financial and non-financial sector*".

**Conflicts of Interest:** The authors declare no conflict of interest.

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
