# Peer review of "Research and Development Efficiency in Public and Private Sectors: An Empirical Analysis of EU Countries by Using DEA Methodology"

_sustainability, doi:10.3390/su12177050_

Round 1
Reviewer 1 Report
The manuscript deals with a very interesting topic, but there are some relatively important issues that need to be considered and corrected.
First of all, the abstract should be shorter, in some places I feel as if I have read selected texts directly from the main text.
Literature review is prepared clearly and uses mostly current literature, which is suitable for the topic. In some places, however, I miss the importance of including Industry 4.0 in some of the theses of the article, because the DEA models that the authors use do not have connections to Industry 4.0 in terms of the variables used.
I am also not very sure whether the variables used are sufficient to evaluate such a large number of countries. The shortcoming of them is the aspect of the quality of research, which is also indicated by the results of efficiency evaluation, where, for example, countries with a high number of highly evaluated universities have lower efficiency scores than countries which do no not have top ranked universities. Their high rating is therefore probably due to low R&D spending. However, this does not imply quality outputs. It is similar in the case of private sector efficiency. Countries that focus more on industry as research are achieving greater efficiency. The chosen variables seem to be too simplified for the given size of the sample of countries. They lack some quality aspects. I also miss the literature sources, especially in section 2.
The inclusion of small countries such as Malta, Luxembourg and Cyprus may also be a problem. These also tend to show high efficiency values ​​in other studies and other areas of research. Subsequently, they affect the results of all other countries. I would consider finding out how this fact can affect the results of efficiency. If the results were significantly different, then it would probably be wise to omit these countries from the research.
I am not sure about the use of DEA models, which assume constant returns to scale. It need to be justified. I miss the literature source that in a given situation, i.e. the use of normalized variables does not recommend the use of variable returns to scale. What about assumption that technology will be more probably based on variable returns to scale?
Some linguistic forms of interpretation are not sufficient for the level of scientific style.
Author Response
Review Report 1
Dear Reviewer,
Thank you very much for the review of our manuscript. We hope that these revisions improve the paper so that you now deem it worthy of publication in "Sustainability ", and also our revision has improved the paper to a level of your satisfaction. We have carefully revised the paper and rewrite it accordingly. Also, we responded in detail to all of the comments. All changes in the revised manuscript are highlighted in yellow.
Thank you for giving us the opportunity to revise our manuscript. We would like to send our revised study again for considering and we look forward to hearing from you.
Kind regards,
Beata Gavurova
Comments and Suggestions for Authors
The manuscript deals with a very interesting topic, but there are some relatively important issues that need to be considered and corrected.
First of all, the abstract should be shorter, in some places I feel as if I have read selected texts directly from the main text.
The abstract was shortened and revised based on the reviewer’s recommendation. It is highlighted in yellow.
Literature review is prepared clearly and uses mostly current literature, which is suitable for the topic. In some places, however, I miss the importance of including Industry 4.0 in some of the theses of the article, because the DEA models that the authors use do not have connections to Industry 4.0 in terms of the variables used.
Literature review that is related to Industry 4.0 (in Introduction and in References) was accordingly amended. It is highligheted in yellow ( lines 46- 56)
While the variables used in the DEA models do not include specific areas of Industry 4.0, also the literature does not specifically define the notion, Industry 4.0 (see Kagermann et al., 2013; Bauer et al., 2014). In this case (based on the selected indicators), the primary focus is on competitiveness, technological changes and innovations that are especially typical of an entrepreneurship in the private sector, which is connected to applied, and /or industrial research and a development of a high-tech technology. As Kagermann, Lukas & Wahlster (2011) state, it is one of the areas of support for Industry 4.0. The source is provided in Introduction and in References.
I am also not very sure whether the variables used are sufficient to evaluate such a large number of countries. The shortcoming of them is the aspect of the quality of research, which is also indicated by the results of efficiency evaluation, where, for example, countries with a high number of highly evaluated universities have lower efficiency scores than countries which do no not have top ranked universities. Their high rating is therefore probably due to low R&D spending. However, this does not imply quality outputs. It is similar in the case of private sector efficiency. Countries that focus more on industry as research are achieving greater efficiency. The chosen variables seem to be too simplified for the given size of the sample of countries. They lack some quality aspects. I also miss the literature sources, especially in section 2.
The number of variables was intentionally decreased in Model 1 (Efficiency R&D in the public sector). In case of ‘the total researchers in the public sector’, there are included 2 indicators (researchers in government sector and researchers in higher education sector) that we cumulatively work with as ‘total researchers in the public sector’. Similarly, in ‘Public expenditures on R&D’, there are used 2 indicators (i.e. government expenditure on R&D and higher education expenditure on R&D) that we cumulatively work with.
In a methodology of Chapter 2, there are amended the output literature sources, and also the methodological procedure is justified in more details. It is highlighted in yellow (lines 219- 222, 231- 233, 241- 247 ). The input and output variables were selected according to the studies mentioned in the literature review (e.g. [5], [26], [34], [35], [39], [40], and [41]). There are standard indicators used in the evaluation of efficiency in the R&D area. From a quality point of view, there is always a question of which indicator is the best one to display the quality of the R&D spending incurred. According to the previous studies, we suppose that number of published scientific papers and citable documents, or a total number of patent applications and high-tech export can adequately describe the quality of R&D spending.
As mentioned by Cook et al. , the selection of inputs and outputs is a very sensitive issue. It is well known that a large number of inputs and outputs compared to the number of DMUs may diminish the discriminatory power of the DEA model. A suggested number of DMUs should be maximum at least three times the number of inputs and outputs combined. As in our analysis, we have 28 countries; it means that the maximum number of variables should be nine. We decided to use not more than five indicators, as other considered indicators published within Eurostat and Scimago had the same informative value.
The next aspect is the orientation of the model. In the input-oriented model, we are looking for countries which use minimum level of inputs to produce a given level of outputs. In the output-oriented model, we are looking for countries that produce the maximum level of outputs with a given level of inputs. In our paper, we apply a non-oriented model, as we capture the desire to improve the input side and output side at the same time and simultaneously. We could not suppose that only reduction in expenditure may help to increase the number of published scientific papers and patents, as also we could not suppose that the given unchanged level of expenditure is the guarantee to increase the number of published scientific papers and patents. The DEA is the method for relative efficiency measurement. It means that the efficiency is calculated in the specified sample and under the given inputs and outputs. Every change in the dataset will lead to different results.
Cook, W. D., Tone, K., & Zhu, J. (2014). Data envelopment analysis: Prior to choosing a model. Omega, 44, 1–4. https://doi.org/10.1016/j.omega.2013.09.004
The inclusion of small countries such as Malta, Luxembourg and Cyprus may also be a problem. These also tend to show high efficiency values ​​in other studies and other areas of research. Subsequently, they affect the results of all other countries. I would consider finding out how this fact can affect the results of efficiency. If the results were significantly different, then it would probably be wise to omit these countries from the research.
In the literature, there are different ways how to eliminate the problem with different size of countries. One of them is to use ratios instead of volume indicators, and the second one is the normalisation. In our paper, we decided to normalise the values of our indicators by using an empirical normalisation. Consequently, it is possible to compare countries with different size, as all input and output values are put into the interval between 0 and 1. This enables us to compare countries with ‘similar size’ (lines 280- 286).
I am not sure about the use of DEA models, which assume constant returns to scale. It need to be justified. I miss the literature source that in a given situation, i.e. the use of normalised variables does not recommend the use of variable returns to scale. What about assumption that technology will be more probably based on variable returns to scale?
The justification is provided in a methodology and also the output literature source. It is highlighted in yellow (292- 298; 300- 312). As mentioned by Jacobs et al. [44], the choice of constant returns to scale or variable returns to scale will usually depend on the context and purpose of the analysis, or whether short-run or long-run efficiency is under scrutiny. For example, from a societal perspective, an interest may be in productivity regardless of the scale of operations so that constant returns to scale may be more appropriate.
The second point is that a complication of the choice of constant returns to scale or variable returns to scale is that data frequently take the form of ratios rather than absolute numbers as measures of inputs and outputs in DEA. It is very common in healthcare or in the case of indicators which are expressed in relation to GDP, which is also the case of expenditures. Therefore, if we have data in the form of ratios, it automatically implies an assumption of constant returns to scale, because the creation of the ratio removes any information about the size of the country. In our sample, we had some data in the form of ratios, but also some data in the form of absolute numbers. Thus, we applied an empirical normalisation to make the same format of the data. The normalised data has a comparable form as the ratios. Consequently, we consider the CRS model as more appropriate.
Some linguistic forms of interpretation are not sufficient for the level of scientific style.
The revision of the article was provided based on the reviewer’s recommendation.
Reviewer 2 Report
The article deals with a very current topic and is based on the original methodological procedure. The aims are clearly defined and the methodology is understandable well to the readers. I recommend to make only minor adjustments:
Introduction: I recommend giving a general definition of “R&D efficiency”.
Materials and methods: I recommend to clearly and verbally define some terms as perceived by the authors of this study: R&D efficiency, technical efficiency, total productivity.
Discussion: In the case of research questions, it is not common to state whether they have been confirmed or verified. These terms are usually connected with hypotheses. The authors should answer yes/no to research questions.
Author Response
Review Report 2
Dear Reviewer,
Thank you very much for the review of our manuscript. We hope that these revisions improve the paper so that you now deem it worthy of publication in "Sustainability ", and also our revision has improved the paper to a level of your satisfaction. We have carefully revised the paper and rewrite it accordingly. Also, we responded in detail to all of the comments. All changes in the revised manuscript are highlighted in yellow.
Thank you for giving us the opportunity to revise our manuscript. We would like to send our revised study again for considering and we look forward to hearing from you.
Kind regards,
Beata Gavurova
Comments and Suggestions for Authors
The article deals with a very current topic and is based on the original methodological procedure. The aims are clearly defined and the methodology is understandable well to the readers. I recommend to make only minor adjustments:
Introduction: I recommend giving a general definition of "R&D efficiency".
General definition of ‘R&D efficiency’ was added in Introduction. It is highlighted in yellow ( lines 89-93)..
Materials and methods: I recommend to clearly and verbally define some terms as perceived by the authors of this study: R&D efficiency, technical efficiency, total productivity.
Economics literature uses a variety of terms to express notions of efficiency and effectiveness, similarly as the literatures of other disciplines. However, these terms are not always defined nor interpreted consistently within and across disciplines. The following terms: R&D efficiency, technical efficiency, and total productivity were added in the following part: Materials and methods (lines 160-169). It is highlighted in yellow.
Discussion: In the case of research questions, it is not common to state whether they have been confirmed or verified. These terms are usually connected with hypotheses. The authors should answer yes/no to research questions.
In the article, the answers to research questions were revised accordingly, YES / NO ( lines 498, 555- 557, 584).
Reviewer 3 Report
- abbreviation DEA should be added in the brackets when using its full name for the first time (line 18); then, explanation of the abbreviation would be unnecessary in line 93;
- statement regarding the diversity of policy priorities across European countries (lines 28-30) lacks any clarification in the main body of the paper, including conclusion;
- the aim of the paper should be clarified – “emphasizing a separate role of public and private sectors” (line 18), “emphasis on a separate role and status of public and private sectors in EU countries” (lines 120-121) are imprecise and lacks any reference in the concluding part; authors assessed public and private sector using separate models, however, did not address the issue of the role or status of each unequivocally, for instance, by stressing the primacy of the private contribution etc.
- also, it is quite confusing, when authors state as follows: “Evaluation of research and development efficiency with an emphasis on a separate role and status of public and private sectors in EU countries is the aim of this study as opposed to numerous comparative analyses and research studies that predominantly evaluate the efficiency of R&D as a whole in all sectors” (lines 120-123) and then, in the next sentence continue: “Also, these consequential facts supported a formulation of the study's aim: evaluation of research and development efficiency in EU countries in public and private sectors and changes of efficiency by using the Malmquist index during 2010/2013 and 2014/2017 based on the empirical analysis” (lines 123-126); aim defined in the first sentence of this paragraph is considered as a “fact” supporting a formulation of the aim in the second sentence, or I missed something?
- literature review addressing previous cross country comparative studies on R&D efficiency using DEA methodology is incomplete – I’d add, among others:
- Abbasi, F., Hajihoseini, H. and Haukka, S. (2010). Use of Virtual Index for Measuring Efficiency of Innovation Systems: A Cross-Country Study, International Journal of Technology Management and Sustainable Development, 9, 195–212
- Cai, Y. (2011). Factors Affecting the Efficiency of the BRICS' National Innovation Systems: A Comparative Study Based on DEA and Panel Data Analysis; Kiel Institute for the World Economy, Kiel
- Dobrzanski, P. (2020). The efficiency of spending on R&D in Latin America region, Applied Economics
- Dobrzanski, P. (2018). Innovation expenditures efficiency in Central and Eastern European Countries. Zbornik Radova Ekonomskog Fakulteta u Rijeci-Proceedings of Rijeka Faculty of Economics, 36(2), 827–859
- Dobrzanski, P. and Bobowski, S. (2020). The Efficiency of R&D Expenditures in ASEAN Countries, Sustainability, 12(7)
- Guan, J.; Chen, K.H. (2012). Modeling the relative efficiency of national innovation systems, Research Policy, 41, 102–115
- Nasierowski,W. and Arcelus, F.J. (2003). On the efficiency of national innovation systems, Socio-Economic Planning Sciences, 37(3), 215–234
In fact, Dobrzanski (2018) can be found among references in the paper, however, not in the context of research using DEA methodology, even though it focused on the European countries too.
- when making a review of previous studies using DEA methodology, authors should point at set of input and output variables selected by authors, a type of DEA model used (CRS input or CRS/VRS output-oriented, super efficiency DEA model, CCR or BCC), list of countries covered by the analysis, as well as key conclusion (if any) for the European countries which are studied in this paper in order to stress its novelty and confront results;
- literature review should also include any reference to EU initiatives aimed at R&D and innovation policies, including EIT, Europe 2020 Strategy (authors just mention this one in line 451) and its flagship initiatives, The Indicator of Innovation Output by the European Commission;
- section 2 “materials and methods” is misleading, as it covers the description of methods only (the first paragraph of this section based on OECD (2015) cannot be considered as a reference to “materials” part. I suggest moving enlarged literature review (in line with suggestions above) to this section instead of keeping it in the introduction part;
- DMU abbreviation lacks explanation in the paper (it was used for the first time in line 159);
- description of methods lacks sufficient references to literature, with special regard to paragraphs with equations; for instance, I don’t know from where Malmquist index equation was obtained as it cannot be fully understood without explanation of variables and symbols used; moreover, instead of t, t+1, authors use 1, 2 which looks like raising variables to the first and second power (page 4-5); I recommend Malmquist index equation from this publication:
- Fare, R., Grosskopf S. (1992). Malmquist Productivity Indexes and Fisher Ideal Indexes. Economic Journal, 158–160
- both in the case of DEA model description in this part of the paper, there is no indication as to the choice of authors (CRS, VRS … etc.); I recommend this publication:
- Simar, L. and Wilson, P. (2002). Nonparametric tests of returns to scale, European Journal of Operational Research, 139, 115–132
- authors calculate the correlation between inputs (or outputs), however, without explanation when they consider it as “high” (line 237-238) – 0.7, 0.8 (R2)? whether any variables were removed from the model as a consequence of that, it should be notified;
- there is no information in the paper what statistical program was used for the calculations; the reviewer is a little bit concerned that without the knowledge about this fact and precise data which were used for calculation correctness of results obtained cannot be verified (some researchers attempt to rely on standard software – in case of which probability of mistake may be higher; if such a situation took place, I suggest to use DEAfrontier Software to make sure that calculations errors do not play a role in the final results of this analysis; the same applies to Malmquist Index Software; to solve this dilemma, I’d suggest to add an appendix with source data used in calculations;
- in case of the Malmquist productivity index, I’d expect an insight into efficiency change (EC) and technical change (TC) components and information as to the method; in this particular case, the authors use the adjacent-base calculation method for Malmquistindex and it is worth mentioning it in the article;
- authors should explain why three-year time lag between input and output was applied – it is relatively long, but maybe results of limited availability of data?
- authors cannot state that country (DMU) is more or less efficient – the country is efficient when reaching 1, inefficient when reaching less than 1;
- the discussion part is quite overloaded with empirical results – numbers and ranks, while lacking proportionate comment from authors;
- lines 548-559 is a kind of review of previous studies on R&D efficiency to mention some variables used – not fully understandable, what is the purpose to do this again just ahead of concluding part of the paper;
- recommendations are pretty limited and overall – “most of the inefficient countries pay attention” (lines 579-580), “In a majority of countries, the next recommendation would be (…)” (lines 582-583) – an added value of this research should be found here, whereas a half of concluding part is a description of the direction of future research and issues related (lines 585-601);
Author Response
Dear Reviewer,
Thank you very much for the review of our manuscript. We hope that these revisions improve the paper so that you now deem it worthy of publication in "Sustainability ", and also our revision has improved the paper to a level of your satisfaction. We have carefully revised the paper and rewrite it accordingly. Also, we responded in detail to all of the comments. All changes in the revised manuscript are highlighted in yellow.
Thank you for giving us the opportunity to revise our manuscript. We would like to send our revised study again for considering and we look forward to hearing from you.
Kind regards,
Beata Gavurova
Comments and Suggestions for Authors
- abbreviation DEA should be added in the brackets when using its full name for the first time (line 18); then, explanation of the abbreviation would be unnecessary in line 93;
- The Abbreviation DEA was added in line 17, other abbreviations (line 133 and 213) were deleted.
- statement regarding the diversity of policy priorities across European countries (lines 28-30) lacks any clarification in the main body of the paper, including conclusion;
In Abstract, the sentence related to ‘the diversity of policy priorities R&D across European countries’ was removed. Also, the paper includes additional connections to ‘policy priorities R&D across European countries and innovation policy’ (Lines 64-69, 100-102, 107-111 or 703- 707)
- the aim of the paper should be clarified – “emphasizing a separate role of public and private sectors” (line 18), “emphasis on a separate role and status of public and private sectors in EU countries” (lines 120-121) are imprecise and lacks any reference in the concluding part; authors assessed public and private sector using separate models, however, did not address the issue of the role or status of each unequivocally, for instance, by stressing the primacy of the private contribution etc.
- also, it is quite confusing, when authors state as follows: “Evaluation of research and development efficiency with an emphasis on a separate role and status of public and private sectors in EU countries is the aim of this study as opposed to numerous comparative analyses and research studies that predominantly evaluate the efficiency of R&D as a whole in all sectors” (lines 120-123) and then, in the next sentence continue: “Also, these consequential facts supported a formulation of the study's aim: evaluation of research and development efficiency in EU countries in public and private sectors and changes of efficiency by using the Malmquist index during 2010/2013 and 2014/2017 based on the empirical analysis” (lines 123-126); aim defined in the first sentence of this paragraph is considered as a “fact” supporting a formulation of the aim in the second sentence, or I missed something?
The authors’ main aim was to evaluate R&D efficiency according to sectors’ performances (in private business sector and in public sector, i.e. government and higher - education sector) and not to evaluate R&D efficiency in all of the sectors as a whole. Thus, an evaluation of R&D efficiency from the perspective of public and private sector provides clearer point of view of scientific research, and innovative activities in the individual countries. The aim’s formulation was revised both in Abstract and in Introduction (Lines 16-17; 186- 191).
- literature review addressing previous cross - country comparative studies on R&D efficiency using DEA methodology is incomplete – I’d add, among others:
- Abbasi, F., Hajihoseini, H. and Haukka, S. (2010). Use of Virtual Index for Measuring Efficiency of Innovation Systems: A Cross-Country Study, International Journal of Technology Management and Sustainable Development, 9, 195–212
- Cai, Y. (2011). Factors Affecting the Efficiency of the BRICS' National Innovation Systems: A Comparative Study Based on DEA and Panel Data Analysis; Kiel Institute for the World Economy, Kiel
- Dobrzanski, P. (2020). The efficiency of spending on R&D in Latin America region, Applied Economics
- Dobrzanski, P. (2018). Innovation expenditures efficiency in Central and Eastern European Countries. Zbornik Radova Ekonomskog Fakulteta u Rijeci-Proceedings of Rijeka Faculty of Economics, 36(2), 827–859
- Dobrzanski, P. and Bobowski, S. (2020). The Efficiency of R&D Expenditures in ASEAN Countries, Sustainability, 12(7)
- Guan, J.; Chen, K.H. (2012). Modeling the relative efficiency of national innovation systems, Research Policy, 41, 102–115
- Nasierowski,W. and Arcelus, F.J. (2003). On the efficiency of national innovation systems, Socio-Economic Planning Sciences, 37(3), 215–234
In fact, Dobrzanski (2018) can be found among references in the paper, however, not in the context of research using DEA methodology, even though it focused on the European countries too.
The authors are aware of the fact that the list of references related to R&D efficiency is definitely not exhausted. Thus, the authors are very grateful and thankful to reviewer for adding other current references relevant to the topic. Also, the authors used other publications rather than those which are related to R&D efficiency with regard to the paper’s focus. As the reviewer recommended, in Literature review’s introduction, there were included some other references (Cai, 2011; Dobrzanski and Bobowski, 2020) (Lines 112-118, 144-154). These references are also a part of the final list of references. The following reference, Dobrzanski (2018), was also added into the part, Methodology, based on the reviewer’s recommendation (Lines 307-308.)
- when making a review of previous studies using DEA methodology, authors should point at set of input and output variables selected by authors, a type of DEA model used (CRS input or CRS/VRS output-oriented, super efficiency DEA model, CCR or BCC), list of countries covered by the analysis, as well as key conclusion (if any) for the European countries which are studied in this paper in order to stress its novelty and confront results;
In Introduction, the review of previous studies by using DEA methodology was expanded with more detailed information. (Lines 164- 185)
- literature review should also include any reference to EU initiatives aimed at R&D and innovation policies, including EIT, Europe 2020 Strategy (authors just mention this one in line 451) and its flagship initiatives, The Indicator of Innovation Output by the European Commission;
Other output documents which are related to EU initiatives and innovation policies were involved within the text and the final list based on the reviewer’s recommendation (Eurostat, 2018, Janger et al., 2017; ITIF, 2019) (Lines 64-69, 100-102, 107-111).
- section 2 “materials and methods” is misleading, as it covers the description of methods only (the first paragraph of this section based on OECD (2015) cannot be considered as a reference to “materials” part. I suggest moving enlarged literature review (in line with suggestions above) to this section instead of keeping it in the introduction part;
The first paragraph in section 2 ‘Materials and methods’ was added to Introduction as a part of Literature review (Lines 72-87).
- DMU abbreviation lacks explanation in the paper (it was used for the first time in line 159);
The abbreviation DMU was explained in line 219.
- description of methods lacks sufficient references to literature, with special regard to paragraphs with equations; for instance, I don’t know from where Malmquist index equation was obtained as it cannot be fully understood without explanation of variables and symbols used; moreover, instead of t, t+1, authors use 1, 2 which looks like raising variables to the first and second power (page 4-5); I recommend Malmquist index equation from this publication:
Fare, R., Grosskopf S. (1992). Malmquist Productivity Indexes and Fisher Ideal Indexes. Economic Journal, 158–160
- The model was updated in formula (3) in line 263 and the source was added to the References.
- both in the case of DEA model description in this part of the paper, there is no indication as to the choice of authors (CRS, VRS … etc.); I recommend this publication:
Simar, L. and Wilson, P. (2002). Nonparametric tests of returns to scale, European Journal of Operational Research, 139, 115–132
- information about how to choose assumption of return to scale was added in lines 222-238 and the source was added to the references
- authors calculate the correlation between inputs (or outputs), however, without explanation when they consider it as “high” (line 237-238) – 0.7, 0.8 (R2)? whether any variables were removed from the model as a consequence of that, it should be notified;
- the information was added to the lines 338-349
- To find out the level of correlation, we apply standard methodology presented by Cohen [24] who classified level of correlation according to the value of the Pearson correlation coefficient. The results could be seen in Table 3a and 3b. According to this classification, we can moderate correlation between inputs and outputs in case of Model 1 except for input, ‘Total researchers in the public sector’, where the correlation between input and both outputs can be considered as very high. The similar situation can be seen in Model 2, where the highest correlation can be observed between ‘Total researchers in the business enterprise sector’ and ‘Patent applications to the EPO’, in other cases the correlation can be considered as moderate. Thus, we can conclude that all our inputs affect at least one output. On the output side, in Model 1, we can see a very high correlation between Scientific and Citable documents, which can signalise that one of these outputs can be considered as unnecessary. Based on the literature review dealing with this topic, we can find both of these outputs as relevant for our analysis. Therefore, we decided not to exclude any of them.
- there is no information in the paper what statistical program was used for the calculations; the reviewer is a little bit concerned that without the knowledge about this fact and precise data which were used for calculation correctness of results obtained cannot be verified (some researchers attempt to rely on standard software – in case of which probability of mistake may be higher; if such a situation took place, I suggest to use DEAfrontier Software to make sure that calculations errors do not play a role in the final results of this analysis; the same applies to Malmquist Index Software; to solve this dilemma, I’d suggest to add an appendix with source data used in calculations;
- line 382-383
- The efficiency score, the Malmquist index and its components were calculated by the mathematical program DEA Solver Pro 13. As all data used in our analysis are available on the web page of Eurostat or Scimago, we decided not to add data used in calculations in the appendix.
- in case of the Malmquist productivity index, I’d expect an insight into efficiency change (EC) and technical change (TC) components and information as to the method; in this particular case, the authors use the adjacent-base calculation method for Malmquistindex and it is worth mentioning it in the article;
- we added formula (2) and description of the catch-up and frontier-shift effect from the theoretical point of view in lines 224 and 265-269
- authors should explain why three-year time lag between input and output was applied – it is relatively long, but maybe results of limited availability of data?
- line 319-323
- we used data for inputs in 2010 and 2014 and for outputs three years later, so in 2013 and 2017. We could not use data for 2018 and 2019 as this data were not available. As it is mentioned in the literature [45], [56] we apply standard methodology for three years lag. The reason is that we assume that in the first year the researchers obtained financial resources from grant and started to work on their research questions. Also, in the second year, they prepared their paper and sent it into the journal. In the last year, the standard processing time is at least one year. Therefore, we suppose that scientific paper will be published in a third year. Consequently, we apply a three-year lag.
- authors cannot state that country (DMU) is more or less efficient – the country is efficient when reaching 1, inefficient when reaching less than 1;
- we made a change in a formulation. Not a country is less or more efficient, but efficiency score is higher or lower compared to other countries.
- the discussion part is quite overloaded with empirical results – numbers and ranks, while lacking proportionate comment from authors;
- lines 548-559 is a kind of review of previous studies on R&D efficiency to mention some variables used – not fully understandable, what is the purpose to do this again just ahead of concluding part of the paper;
The purpose was to support our findings of R&D efficiency by also other researches and studies that applied similar indicators, and subsequently, to point to the achieved results.
- recommendations are pretty limited and overall – “most of the inefficient countries pay attention” (lines 579-580), “In a majority of countries, the next recommendation would be (…)” (lines 582-583) – an added value of this research should be found here, whereas a half of concluding part is a description of the direction of future research and issues related (lines 585-601);
In Conclusion, there were included recommendations and also the direction of future research was slightly revised (lines 703- 707; 711- 719).
Round 2
Reviewer 1 Report
Accept in present form
Reviewer 3 Report
I found revised paper as valuable contribution to studies on R&D efficiency using DEA methodology. Author followed recommendations of reviewer and improved the paper.
I suggest to check the language of the paper, as some new parts added to the paper in line with suggestions of reviewer involve some spelling mistakes.